# The Protective Effect of Zebularine, an Inhibitor of DNA Methyltransferase, on Renal Tubulointerstitial Inflammation and Fibrosis

**DOI:** 10.3390/ijms232214045

**Published:** 2022-11-14

**Authors:** Eun Sil Koh, Soojeong Kim, Mina Son, Ji-Young Park, Jaehyuk Pyo, Wan-Young Kim, Minyoung Kim, Sungjin Chung, Cheol Whee Park, Ho-Shik Kim, Seok Joon Shin

**Affiliations:** 1Department of Internal Medicine, College of Medicine, The Catholic University of Korea, Seoul 06591, Republic of Korea; 2Department of Biochemistry, College of Medicine, The Catholic University of Korea, Seoul 06591, Republic of Korea; 3Department of Biomedicine and Health Sciences, College of Medicine, The Catholic University of Korea, Seoul 06591, Republic of Korea; 4Division of Nephrology, Department of Internal Medicine, Incheon St. Mary’s Hospital, College of Medicine, The Catholic University of Korea, Incheon 21431, Republic of Korea

**Keywords:** inflammation, fibrosis, DNA methyltransferase, unilateral ureteral obstruction, oxidative stress

## Abstract

Renal fibrosis, the final pathway of chronic kidney disease, is caused by genetic and epigenetic mechanisms. Although DNA methylation has drawn attention as a developing mechanism of renal fibrosis, its contribution to renal fibrosis has not been clarified. To address this issue, the effect of zebularine, a DNA methyltransferase inhibitor, on renal inflammation and fibrosis in the murine unilateral ureteral obstruction (UUO) model was analyzed. Zebularine significantly attenuated renal tubulointerstitial fibrosis and inflammation. Zebularine decreased trichrome, α-smooth muscle actin, collagen IV, and transforming growth factor-β1 staining by 56.2%. 21.3%, 30.3%, and 29.9%, respectively, at 3 days, and by 54.6%, 41.9%, 45.9%, and 61.7%, respectively, at 7 days after UUO. Zebularine downregulated mRNA expression levels of *matrix metalloproteinase* (*MMP*)-*2*, *MMP-9*, *fibronectin*, and *Snail1* by 48.6%. 71.4%, 31.8%, and 42.4%, respectively, at 7 days after UUO. Zebularine also suppressed the activation of nuclear factor-κB (NF-κB) and the expression of pro-inflammatory cytokines, including *tumor necrosis factor-α*, *interleukin* (*IL*)-*1β*, and *IL-6*, by 69.8%, 74.9%, and 69.6%, respectively, in obstructed kidneys. Furthermore, inhibiting DNA methyltransferase buttressed the nuclear expression of nuclear factor (erythroid-derived 2)-like factor 2, which upregulated downstream effectors such as catalase (1.838-fold increase at 7 days, *p* < 0.01), superoxide dismutase 1 (1.494-fold increase at 7 days, *p* < 0.05), and NAD(P)H: quinone oxidoreduate-1 (1.376-fold increase at 7 days, *p* < 0.05) in obstructed kidneys. Collectively, these findings suggest that inhibiting DNA methylation restores the disrupted balance between pro-inflammatory and anti-inflammatory pathways to alleviate renal inflammation and fibrosis. Therefore, these results highlight the possibility of DNA methyltransferases as therapeutic targets for treating renal inflammation and fibrosis.

## 1. Introduction

End-stage renal disease (ESRD), the final phase of chronic kidney disease (CKD), is a growing public health burden, and thus intensive efforts have been made worldwide to prevent or delay the progression of CKD [1]. Renal fibrosis is the hallmark of most progressive CKD and represents the final, common, and typical response to renal injury [2]. From the viewpoint of developing mechanisms, renal fibrosis is coordinated by a complex network of epithelial-mesenchymal transition (EMT), inflammation, and oxidative stress [3,4]. Epigenetic mechanisms further intensify the developmental complexity of CKD. Thus, CKD patients with identical etiological factors have different clinical outcomes of renal fibrosis, due to genetic and epigenetic differences [5].

Epigenetic modifications primarily consist of three basic processes: methylation of the DNA, modification of the histone proteins, and the regulation of gene expression by non-coding RNAs [6]. DNA methylation occurs at the 5′-position of cytosine in CpG dinucleotides through the action of DNA methyltransferase (DNMT), which catalyzes the transfer of methyl groups from S-adenosyl methionine (SAM) [7]. Most CpG dinucleotides are grouped in clusters at the 5′-regulatory regions of many genes [8]. Compared to genomic DNA sequence averages of 40% GC content, the GC content of this region reaches up to 60%, and are thereby called CpG islands [7,8]. The DNA methylation of promoter CpG islands generally transforms euchromatin to heterochromatin, which suppresses transcription initiation and leaky transcription [7,8]. Zebularine, 1-(b-D-ribofuranosyl)-1,2-dihydropyrimidin-2-one, is a structural analog of cytidine and an inhibitor of DNMT [9]. As a cytidine analog, zebularine incorporates into DNA and binds to DNMT to trap it by suppressing its enzymatic activity [10,11]. Zebularine was shown to reverse the silent expression of tumor suppressor genes, including *p16* (*INK4A*), which is hypermethylated in cancer cells [12]. Compared to other DNMT inhibitors, such as 5′-azacytidine and 5′-azadeoxycytidine, zebularine is more stable, is orally administered, and is less toxic to normal cells and tissues, making it suitable to apply in animal experiments [13].

The contribution of DNA methylation to the process of renal fibrosis has received much attention from researchers, but has not yet been fully elucidated. Recent studies have demonstrated the correlation between the methylation status of genomic DNA and the severity of diabetic kidney disease (DKD), suggesting the potential role of DNA methylation in DKD [14,15]. For instance, pathologic findings, such as albuminuria, podocyte injury, mesangial matrix expansion, and glomerular hypertrophy, were alleviated in a DKD mouse model after treatment with 5′-azacytidine, an inhibitor of DNA methylation [16]. Another study reported that the DNMT1-mediated hypermethylation of the *RASAL1* gene was associated with the activation of fibroblasts and fibrogenesis in the kidney [17]. It should be noted that the expression of DNMT1 was augmented in diabetic kidneys and attenuated by the treatment with 5′-azacytidine, implying that the DNA methyltransferase was a therapeutic target against DKD [16,17]. However, little is known about the involvement of DNA methylation in the pathogenesis of non-diabetic renal fibrosis. Here, we hypothesized that unbalanced DNA methylation is a molecular cause of progressive fibrosis. Thus, we examined the effect of zebularine, a DNMT inhibitor, on the progression of EMT, inflammation, and oxidative stress in murine kidneys with unilateral ureteral obstruction (UUO).

## 2. Results

### 2.1. Expression of DNMT in Obstructed Kidneys

Immunohistochemical staining for DNMT1 showed a marked increase in the tubulointerstitium of kidneys on days 3 and 7 after UUO compared with that in the kidneys of sham-operated mice (Figure 1A,B). However, the DNMT inhibitor zebularine attenuated the increased staining to a degree similar to that in sham-operated mice. We examined renal protein levels and relative mRNA expression of DNMT in UUO mice. DNMT1 and DNMT3b protein levels increased on days 3 and 7 post-UUO, and consistently with the previous reports, zebularine suppressed the increase (Figure 1C–E). The zebularine treatment also reduced the relative mRNA expression of DNMT1 and DNMT3b (Figure 1F,G).

### 2.2. Effects of DNMT Inhibition on Fibrotic Changes in Obstructed Kidneys

Renal tubulointerstitial fibrosis gradually appeared to be apparent after UUO, as shown by Masson’s trichrome staining, when compared with the sham-operated mice (Figure 2A). Notably, zebularine significantly reduced the area of fibrotic lesions in obstructed kidneys. Concurrently, a significant increase in total collagen content and type IV collagen expression was detected in UUO kidneys, but both were effectively attenuated by zebularine treatment (Figure 2B,C). The protein level of transforming growth factor (TGF)-β1, the principal profibrotic cytokine, also increased in obstructed kidneys. However, the administration of zebularine significantly suppressed the increase in TGF-β1 protein levels (Figure 2D). In addition, relative *TGF-β1* mRNA expression increased markedly in obstructed kidneys, which was also attenuated by zebularine (Figure 2E). These results suggest that DNA methylation under a UUO-induced renal insult is associated with the development and progression of kidney fibrosis, and that inhibiting DNA methylation has a beneficial effect on renal fibrosis after obstructive injury.

### 2.3. Effects of DNMT Inhibition on Epithelial-Mesenchymal Transition (EMT) in Obstructed Kidneys

Then, we examined the expression of α-smooth muscle actin (α-SMA), a relevant marker of myofibroblast activation, and EMT-related genes to investigate the effect of DNMT on EMT during renal fibrosis development. As shown in Figure 3A, zebularine treatment significantly reduced immunohistochemical staining for α-SMA in obstructed kidneys on days 3 and 7. Zebularine treatment also resulted in a strong decrease in the relative mRNA expression levels of *vimentin*, *fibronectin*, *MMP-2*, *MMP-9*, and *N-cadherin* in obstructed kidneys (Figure 3B–G). Moreover, renal protein and relative mRNA expression levels of *Snail1*, a key regulator of EMT, increased in obstructed kidneys and were attenuated by zebularine (Figure 3H–J). All these results suggest that the inhibition of DNMT by zebularine prevents EMT during UUO-induced development of renal fibrosis.

### 2.4. Inhibition of DNMT Attenuates Inflammation in Obstructed Kidneys

Since inflammation is one of the key contributors to the development of renal fibrosis, we performed immunohistochemistry of F4/80-positive cells in the tubulointerstitium of the obstructed kidney to investigate the effect of DNMT inhibition on renal inflammation. As shown in Figure 4A, the number of F4/80-positive cells increased in obstructed kidneys, but shrunk significantly in zebularine-treated kidneys. Relative mRNA expression levels of pro-inflammatory cytokines, including *interleukin* (*IL*)-*1β*, *IL-6*, *IL-17*, and *tumor necrosis factor* (*TNF*)-*α*, were enhanced at three and seven days post-UUO (Figure 4B–E). However, their expression was also significantly suppressed by zebularine treatment. Moreover, the increase in nuclear NF-κB (p65), a major inflammatory response regulator in renal fibrosis, post-UUO was significantly attenuated by zebularine treatment (Figure 4F). Collectively, these findings indicate that DNMT inhibition by zebularine treatment alleviated the activation of NF-κB and the subsequent inflammatory pathway in obstructed kidneys, which supports the mechanism for the anti-inflammatory effect of zebularine on renal fibrosis.

### 2.5. Inhibition of DNMT Decreases Oxidative Stress and Increases Antioxidant Enzymes in Obstructed Kidneys

Oxidative stress is a well-known cause of renal inflammation in the UUO model. Thus, we analyzed the effect of zebularine on UUO-induced oxidative stress. The protein and mRNA levels of Nox2, a primary source of reactive oxygen species (ROS), increased markedly in obstructed kidneys, which was prevented in zebularine-treated kidneys (Figure 5A–C). After UUO, the obstructed kidneys exhibited suppressed expression of antioxidant defense and cytoprotective proteins, such as catalase and superoxide dismutase 1 (SOD1). Zebularine significantly increased the expression of these proteins (Figure 5F–I). Interestingly, NAD(P)H: quinone oxidoreductase 1 (Nqo1) increased after the obstructive injury, and zebularine further boosted its expression (Figure 5D,E). Nuclear factor (erythroid-2)-related factor 2 (Nrf2) has a critical role in coordinating the induction of several genes encoding antioxidant molecules [18]. The expression of nuclear Nrf2 was reduced in response to UUO, but reversed significantly by zebularine treatment (Figure 5J,K). Accordingly, these results indicate that inhibition of DNMT by zebularine may prevent the UUO-induced inhibition of Nrf2 activity and its downstream antioxidant pathway. Interestingly, U3Ze showed a more substantial inhibitory effect than U7Ze. Zebularine almost completely prevented the up-regulation of Nox2 expression at three days in obstructed kidneys. In addition, the decrease in nuclear Nrf2 and the increase in Nqo1 were prominent at three days post-UUO. However, as oxidative stress sustained beyond three days until seven days, the effect of zebularine seemed to weaken gradually. Therefore, it was likely that U3Ze showed a better effect than U7Ze.

### 2.6. Effects of DNMT Inhibition on Apoptosis in Obstructed Kidneys

Cell death by apoptosis is also involved in renal fibrosis. Thus, we examined apoptotic cells by a terminal deoxynucleotidyl transferase dUTP nick-end labelling (TUNEL) assay, as well as the expression of the pro-apoptotic protein Bax and the anti-apoptotic protein Bcl-2 by immunoblot analysis, to investigate the effects of zebularine on apoptosis in UUO kidneys. As shown in Figure 6A, the number of TUNEL-positive cells gradually increased with time, and zebularine conspicuously reversed this effect in the obstructed kidney. Consistent with the induction of apoptosis, UUO caused a decline in the Bcl-2 to Bax ratio, which was also prevented significantly by zebularine treatment (Figure 6B,C). These results suggest that DNMT inhibition by zebularine protects the kidney from UUO-induced apoptosis.

## 3. Discussion

The present results demonstrate that the treatment of zebularine, an inhibitor of DNMT, ameliorated renal inflammation and fibrosis in the mouse UUO model, which is an established model of non-metabolic renal fibrosis. Improved kidney fibrosis was accompanied by attenuated EMT and apoptosis in response to UUO. Thus, the present study suggests that DNMT participates in inflammation, EMT, and apoptosis, all of which are involved in the pathogenic process of renal fibrosis.

Regarding inflammation, zebularine treatment alleviated the activation of NF-κB and the subsequent expression of inflammatory cytokines in obstructed kidneys (Figure 4). In the UUO model, inflammatory cytokines increased, which was followed by the expression of anti-inflammatory or pro-fibrotic cytokines [19]. The intriguing part of the results is that zebularine also attenuated anti-inflammatory molecules, such as IL-10 and IL-11, as well as pro-inflammatory cytokines (Appendix A). So, it can be inferred that DNMT may affect the expression of most cytokines, regardless of their characteristics and roles, and both pro-inflammatory and anti-inflammatory cytokines may be targets of zebularine, which prevents the expression of cytokines in a mechanistically specific manner. Given that the final effect of inhibiting DNMT during renal fibrosis was more favorable for renoprotection, DNMT appears to be more influential toward pro-inflammatory than anti-inflammatory conditions in the mouse UUO model. Accordingly, this observation suggests that inhibiting DNMT in renal fibrosis could be a therapeutic strategy to correct the imbalance between pro-inflammatory and anti-inflammatory molecules.

Previous studies have primarily focused on the associations between metastatic cancer and EMT along with inhibiting DNMT [20,21]. The hypermethylation of tumor suppressor gene promoters has been observed. Increased promoter methylation interferes with transcription factor binding, resulting in the loss of tumor suppressor expression, which thereby contributes to malignant transformation [22]. EMT also has been thought to be an adaptive response to chronic injury that participates in the development of renal fibrosis [23,24]. During EMT, renal tubular epithelial cells lose their epithelial characteristics, as evidenced by a decrease in the concentration of adhesion molecules, and simultaneously acquire migratory and mesenchymal phenotype producing mesenchymal gene products, such as vimentin, fibronectin, MMP-2, and MMP-9 [25]. As described in previous reports, Snail1, which is involved in cell differentiation and survival, has a pivotal role in regulating EMT as a transcriptional regulator [26,27]. Notably, NF-κB also regulates *Snail1*, and in a previous report, Snail1-induced EMT was not reversed by DNMT inhibitors [28]. However, in our results, inhibiting DNMT with zebularine debilitated the Snail1-related EMT process (Figure 3). Zebularine treatment suppressed the expression of nuclear NF-κB (Figure 4F) and Snail1 (Figure 3H), leading to the blockage of EMT and renal fibrosis. Based on these findings and previous reports, we could speculate that the primary cause of EMT inhibition by zebularine treatment may be the prevention of NF-κB activation. In other words, zebularine treatment prevents NF-κB activation and inhibits the induction of its target gene *Snail1*, which then cannot induce the expression of EMT-related genes. Therefore, EMT appears to be subjected to epigenetic regulation and to be prevented or reversed by inhibiting DNMT during renal fibrosis. The renoprotective effect of zebularine raises the possibility of aberrant DNA hypermethylation in obstructed kidneys. Since DNMT1 and DNMT3b were highly expressed by UUO operation (Figure 1A–C), we postulated that aberrant DNA hypermethylation is caused by the overexpression of DNMT with increased enzymatic activity. Micevic et al. proposed that DNA hypermethylation could also be caused by the aberrant targeting of DNMT, such as altered splicing, altered binding, or altered expression of scaffolding proteins [29]. So, there may be diverse mechanisms or pathways involved in UUO-induced DNA hypermethylation, which should be identified for the new development of therapeutics exploiting DNA methylation against renal fibrosis.

DNMT has three bioactive forms: DNMT1, DNMT3a, and DNMT3b. The maintenance of DNA methylation in somatic cells is generally promoted by DNMT1, and de novo DNA methylation occurs through DNMT3a and DNMT3b [30]. However, overlap in the function of these two DNMTs occurs in some areas of the genome. Our findings revealed the consistent results of each DNMT isoform; the renal protein level and mRNA expression of DNMT isoforms increased after the UUO operation but decreased after zebularine treatment. An indirect process facilitated by other epigenetic modifications could regulate DNA methylation status. Yin et al. demonstrated that fibrotic kidneys displayed marked promoter hypermethylation of the *Klotho* gene, which was mediated by activating DNMT1/3a, and resulted in the suppressed expression of *Klotho*. Interestingly, the expression of DNMT1/3a was facilitated by TGFβ-1-induced downregulation of miR-152/30a [31]. Therefore, further research about complex genetic and epigenetic interactions leading to the increased expression of DNMT should be performed to understand the mechanism involved in the renoprotective effect of zebularine.

It is well known that the balance between reactive oxygen species (ROS) production and ROS scavenging is a crucial modulator in the progression of renal fibrosis. In this study, we evaluated the expression of *Nox2*, which tended to increase in obstructive kidneys, whereas zebularine treatment reversed its increase. Furthermore, it was accompanied by the restoration of anti-oxidative defense proteins, such as Nqo1, catalase, and SOD1, of which transcription was stimulated by Nrf2. Nrf2 has been shown to function as an anti-inflammatory modulator via the negative regulation of NF-κB [32,33,34]. The antioxidant effect of Nrf2 makes an intracellular environment that favors the reduced or inactive form of NF-κB [35]. Mechanically, Nrf2 prevents the phosphorylation and degradation of IκB-α, as well as thesubsequent nuclear translocation of NF-κB [36]. In this study, inhibiting DNMT with zebularine preserved the nuclear expression of Nrf2 with an increase in the subsequent anti-oxidative defense proteins. Zebularine treatment also repressed the accumulation of nuclear NF-κB, which was consistent with the negative effect of Nrf2 on NF-κB nuclear translocation. These results support the proposition that inhibiting DNMT may contribute to the reno-protective effect by regulating the NF-κB and Nrf2 pathways. Interestingly, NF-κB has been shown to interact with DNMT1 to repress the transcription of its target genes [37]. Therefore, although the molecular mechanism involved in the reno-protective effect of zebularine in the UUO model was not clarified in detail in this study, the Nrf2 recovery and inhibition of NF-κB activation may be the critical event.

The three major targets of zebularine in the UUO model may be Nrf2, NF-κB, and Nox2. By modulating these molecules, zebularine suppressed oxidative stress, inflammation, and EMT, which led to the prevention of fibrosis in UUO (Appendix A).

This study has the following limitations. The first limitation of this study is that the side effect of zebularine, such as hepatic injury, was not measured. The first-pass metabolism of zebularine occurs in the liver. Zebularine is metabolized to uridine by aldehyde oxidase (AO), which is abundant in hepatocyte cytosol [38]. Since the first-pass metabolic rate of zebularine is somewhat rapid, its oral bioavailability seems to be limited and safe, with less side effects than that of other demethylating agents, such as 5-azacytidine and 5-azadeoxycytidine [39]. To be consistent, mice were well tolerable to zebularine administered intraperitoneally or intravenously, and showed no sign of liver or kidney injury [13,40,41]. Moreover, diabetic rats treated intraperitoneally with zebularine at a dose of 225 mg/kg/day daily for 14 days were tolerable and prolonged the survival of pancreatic islet allotransplants for up to 90 days [42]. Therefore, mice in this study treated with zebularine at a dose of 225 mg/kg/day daily for three or seven days should have been well tolerable, with no hepatic or kidney injuries. However, since the administration of zebularine by daily intravenous injection at a dose of 250 mg/kg/day for 10 days with a 2-day interval induced a modest increase in the hepatic enzyme ALT in cynomolgus monkeys [43], it should be considered that the safety of zebularine treatment is variable among species. The second limitation of this study is that the effect of zebularine on the renal function was not analyzed. Since many previous studies have reported that BUN or serum creatinine was not significantly changed by UUO, because the contralateral kidney had good function [44,45] and usually compensated for the decreased function of the UUO kidney [46], BUN and serum creatinine do not seem to precisely reflect the renal function in an animal model of UUO [47]. So, it is expected that the effect of zebularine on the renal function may be obvious in a model where both kidneys undergo fibrosis. The third limitation of this study is that the effect of the metabolites of zebularine was not analyzed. It is known that zebularine is metabolized sequentially to uridine, uracil, dihydrouracil, β-ureiodopropionic acid and β-alanine [39]. Since the half-life of zebularine is short, its bioavailability is limited and safe for animals with less side effects. Although pyrimidine analogs have hepatotoxic effects, metabolites of zebularine are indistinguishable from endogenous ones, and don’t seem to have toxic effects. It was reported that, in patients with diabetic kidney disease, dihydrouracil and β-ureidopropionic acid were significantly down-regulated, which led to a decrease in pantothenate and CoA biosynthesis [48]. Therefore, it can be speculated that zebularine might have therapeutic potential in diabetic kidney disease.

In conclusion, inhibiting DNMT ameliorated renal fibrosis in obstructed kidneys. Our findings suggest that inhibiting DNA methylation restrains unnecessarily intensified pro-inflammatory pathways, restores disrupted anti-oxidative defense mechanisms, and alleviates EMT and apoptosis, thus ameliorating renal fibrosis. Hence, DNMT may be a valid therapeutic target to protect against renal fibrosis in non-diabetic CKD.

## 4. Materials and Methods

### 4.1. Animal Experiment

All animal experiments were performed following the relevant guidelines and regulations as described in the previous report [49]. Briefly, male C57BL/6 mice (weight 20–25 g; OrientBio, Inc., Seoul, Republic of Korea) underwent left ureteral ligation with 4-0 silk thread under general anesthesia; the sham operation was performed similarly without ligation. Mice received an intraperitoneal injection of 225 mg/kg/day of zebularine (Tokyo Chemical Industry Co., Ltd., Tokyo, Japan) or 200 μL vehicle alone, beginning on day 1 for three or seven days after the operation. The dose of zebularine chosen in this study was expected to be non-toxic based on the previous report [42]. As shown in Figure 7, the mice were divided into five groups (n = 6 per group): sham-operated control mice, control mice sacrificed on day 3 after UUO, zebularine-treated mice sacrificed on day 3 after UUO, control mice sacrificed on day 7 after UUO, and zebularine-treated mice sacrificed on day 7 after UUO. At the end of the experiment, the kidneys were harvested for histological evaluation and molecular measurements. All experiments were performed under protocols approved by the Institutional Animal Care and Use Committee of The Catholic University of Korea, Yeouido St. Mary’s Hospital (No. YEO20161602FA).

### 4.2. Histology and Immunohistochemistry (IHC)

Histology and immunohistochemistry experiments were performed following the previous reports [50,51]. Briefly describing, phosphate-buffered paraformaldehyde (4%)-fixed tissue sections were stained with Masson’s trichrome to evaluate the severity of tubulointerstitial fibrosis in the kidney. More than 6 fields per kidney tissue, with a total of 40 fields per experimental group, were randomly chosen and photographed manually, and the fibrotic area was detected and quantified using ImageJ 1.49 software (National Institutes of Health, Bethesda, MD, USA). The difference among experimental groups was statistically verified using one-way analysis of variance with Bonferroni correction.

We conducted immunohistochemical staining for DNMT1 (Cell Signaling Techology, Denvers, MA, USA) to investigate the effect of zebularine on DNMT1 expression. To determine the number of myofibroblasts or macrophages, sections were incubated with anti-α-SMA or anti-F/80 antibodies (Abcam, Cambridge, UK), respectively. After washing in PBS, all sections were incubated for 60 min with peroxidase-conjugated anti-mouse IgG (Jackson ImmunoResearch Laboratories, West Grove, PA, USA) as a secondary antibody, and then reacted with a mixture of 3,3′-diaminobenzidine (0.05%)-containing H_2_O_2_ (0.01%) for the color reactions. After counterstaining with haematoxylin, more than 6 fields per kidney tissue, with total 40 fields per experimental group, were randomly selected and photographed manually. Stain-positive cells were detected and quantitated by ImageJ software 1.49 (National Institutes of Health). To assess apoptosis, the TUNEL assay was performed according to the manufacturer’s protocol (Millipore, Billerica, MA, USA) and a previous report [39]. TUNEL-positive cells were evaluated in 40 randomly selected tubulointerstitial fields per section using ImageJ 1.49 software (National Institutes of Health). All slides were assessed in a blind manner.

To convince the integrity of IHC procedure, IHC for negative images were performed. To include the negative control, we conducted IHC against U7C, which showed the highest positivity, by skipping primary antibody incubation. Briefly, a section of U7C was incubated in primary antibody incubating solution without primary antibodies. After washing in PBS, the section was incubated for 60 min with peroxidase-conjugated anti-rat or rabbit or mouse IgG (Jackson ImmunoResearch Laboratories) as a secondary antibody, and then reacted with a mixture of 3,3′-diaminobenzidine (0.05%)-containing H_2_O_2_ (0.01%) for the color reaction. After counterstaining with haematoxylin, the section was photographed (Appendix A).

### 4.3. Renal Collagen Content Assay

The total collagen content of the kidney tissue was measured using the acid hydrolysis method following previous reports [47,52]. Briefly describing, each kidney tissue was hydrolyzed in HCl (6 N) for 18 h at 110 °C and dried at 75 °C. After the samples were solubilized in citric acid collagen buffer, they were filtered through centrifugal filter units (EMD Millipore, Darmstadt, Germany) and oxidized with a chloramine-T solution. Ehrlich’s reagent (100 μL) was then added to start the color reaction. The absorbance of samples was measured at 550 nm. The total collagen in kidney tissue was calculated based on the assumption that collagen contains 12.7% hydroxyproline by weight.

### 4.4. Immunoblot Analysis

The total proteins in kidney tissues were extracted using the PRO-PREP Protein Extraction Kit (iNtRON Biotechnology, Seongnam-si, Gyeonggi-do, Republic of Korea), and nuclear proteins in kidney tissues were extracted using the NE-PER Nuclear and Cytoplasmic Extraction Kit (Thermo Fisher Scientific, Waltham, MA, USA) according to the manufacturer’s instructions. Protein concentrations were determined using a protein assay kit (Bio-Rad Laboratories, Hercules, CA, USA). After electrophoresis, the proteins in the gel were transferred to a nitrocellulose membrane. The membrane was incubated overnight at 4°C with primary antibodies against the following proteins: collagen IV, catalase (Abcam), TGF-β1, NF-κB, NQO1, nuclear Nrf2, Bcl-2, Bax, DNMT1, DNMT 3b (Santa Cruz Biotechnology, Dallas, TX, USA), Nox2 (BD Biosciences, San Jose, CA, USA), HO-1 (Thermo Fisher Scientific), SOD1 (Enzo Life Science, Inc., Farmingdale, NY, USA), β-actin (Sigma-Aldrich, St. Louis, MO, USA), and Lamin B1 (Cell Signaling Technology). After a wash, the blots were incubated with a secondary antibody conjugated with horseradish peroxidase. The protein bands were detected with enhanced chemiluminescence reagents and imaged using an Image Quant LAS 4000 (GE Healthcare, Piscataway, NJ, USA). Their intensities were measured by Quantity One 1-D analysis software (Bio-Rad Laboratories), and normalized by β-actin or Lamin B1 in the same sample.

### 4.5. Quantitative Real-Time Polymerase Chain Reaction (qRT-PCR)

The total RNA was isolated from kidney tissues using TRIzol Reagent (Thermo Fisher Scientific) according to the manufacturer’s manual. The reverse transcriptase reaction was performed to synthesize cDNA, and qRT-PCR assays were performed using SYBR Premix (Takara Bio, Inc., Otsu, Japan). Primer sequences for each gene are listed in the Appendix A. The specificity of the PCR products was confirmed by analyzing the melting curves. All PCRs were performed in duplicate and repeated independently three times. The expression level of GAPDH mRNA normalized the mRNA level of each gene in the same sample, and their relative changes among samples were calculated by the ∆∆Ct method [53].

### 4.6. Statistical Analysis

All values in this study are represented as the mean ± standard error of the mean (SEM). Differences between groups were determined using one-way analysis of variance with Bonferroni correction.

## Figures and Tables

**Figure 1 ijms-23-14045-f001:**
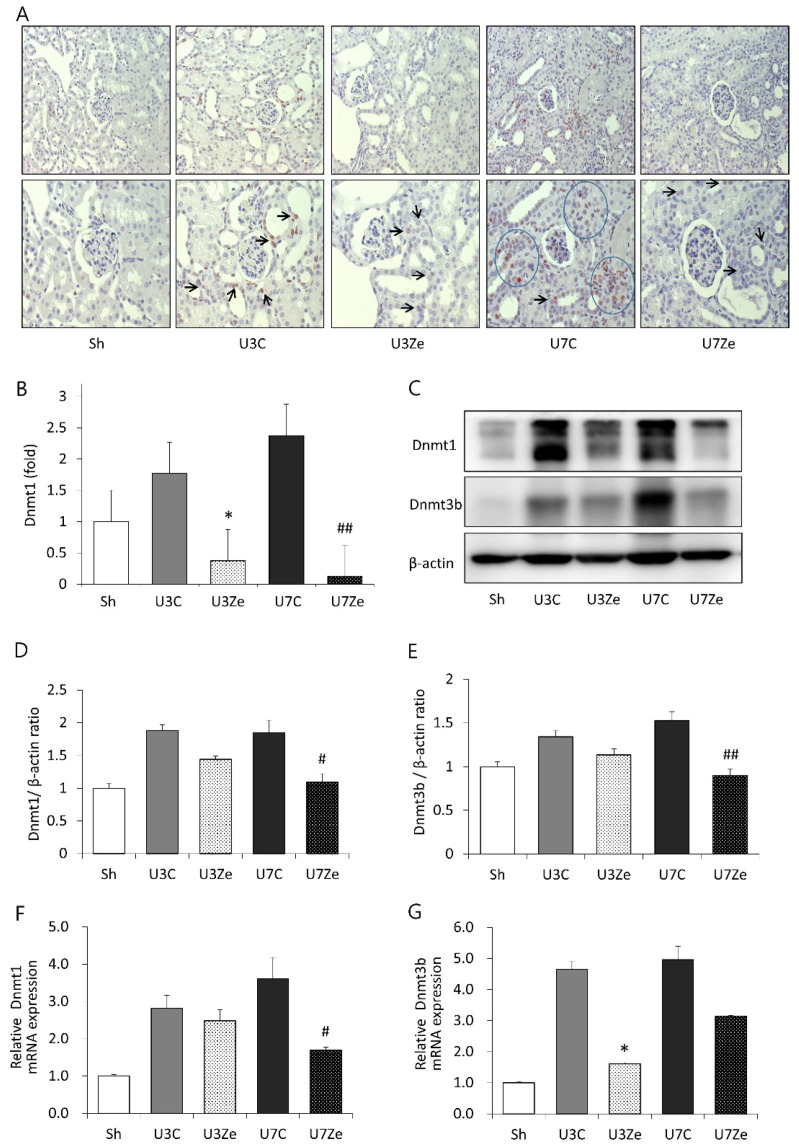
Expression of DNA methyltransferases in obstructed kidneys. (**A**,**B**) Representative renal sections stained with DNMT1 (original magnification: upper, ×200; lower, ×400) and quantitative analyses. Arrows and circles indicate positive cells and areas, respectively. (**C**–**E**) Western blot and quantitative analyses of DNMT1 and DNMT3b. (**F**,**G**) QRT-PCR analysis for DNMT1 and DNMT3b in obstructed kidney tissue after inhibiting DNMT. * *p* < 0.05 vs. UUO D3 control; ^#^
*p* < 0.05 vs. UUO D7 control; ^##^
*p* < 0.01 vs. UUO D7 control. Sh, sham-operated group; U3C, UUO at day 3 group; U3Ze, zebularine-treated UUO group at day 3; U7, UUO at day 7 group; U7Ze, zebularine-treated UUO group at day 7.

**Figure 2 ijms-23-14045-f002:**
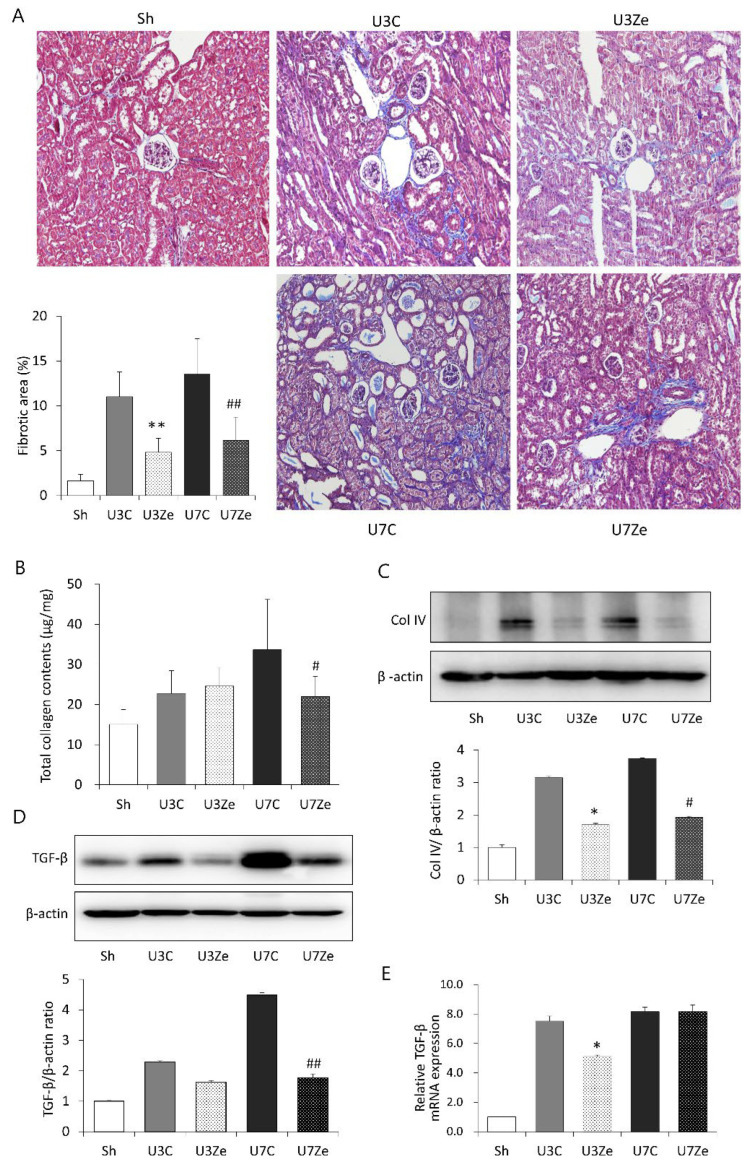
The effect of zebularine on renal tubulointerstitial fibrosis after UUO. (**A**) Representative renal section stained with Masson’s trichrome (original magnification, ×200) and quantitative analyses of the results of tubulointerstitial fibrosis area. (**B**) Change in total collagen content after zebularine treatment in obstructive kidneys after UUO. (**C**,**D**) Representative immunoblot images showing that renal type IV collagen (Col IV) and TGF-β1 expression decreased in the zebularine group. Density of each protein band was normalized by β-actin of the same sample. (**E**) TGF-β1 mRNA level was attenuated in obstructed kidneys by zebularine. Expression level of TGF-β1 mRNA in each sample was normalized by the expression level of GAPDH mRNA in the same sample. * *p* < 0.05 vs. UUO D3 control; ** *p* < 0.01 vs. UUO D3 control; ^#^
*p* < 0.05 vs. UUO D7 control; ^##^
*p* < 0.01 vs. UUO D7 control. Sh, sham-operated group; U3C, UUO at day 3 group; U3Ze, zebularine-treated UUO group at day 3; U7, UUO at day 7 group; U7Ze, zebularine-treated UUO group at day 7.

**Figure 3 ijms-23-14045-f003:**
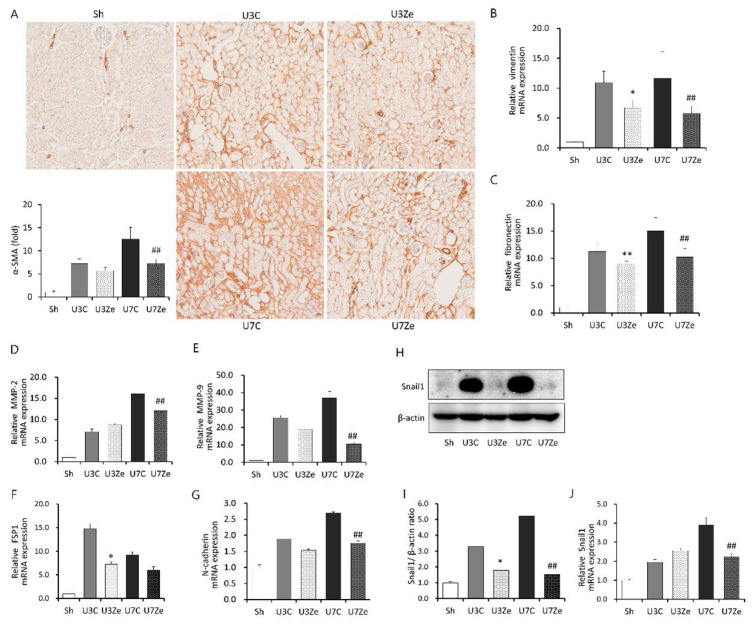
The effect of zebularine on renal expression of EMT-related genes after UUO. (**A**) Representative renal sections stained with α-SMA (original magnification, ×200) and quantitative analyses. (**B**–**G**) QRT-PCR analyses of the obstructed kidney tissues showing upregulation of *vimentin* (**B**), *fibronectin* (**C**), *matrix metalloproteinase* (*MMP*)-2 (**D**), *MMP*-9 (**E**), *fibroblast stimulating protein* (*FSP*)1 (**F**), and *N-cadherin* (**G**), whose expression changed significantly in response to zebularine on day 3 or 7 or both after UUO. (**H**,**I**) Representative immunoblot image against Snail1 (**H**) and densitometric results of immunoblot images (**I**). Intensities of Snail1 bands were normalized by the β-actin in the same sample. (**J**) QRT-PCR result of *Snail1* mRNA. The expression level of *Snail1* mRNA was normalized by the *GAPDH* mRNA in the same sample. * *p* < 0.05 vs. UUO D3 control; ** *p* < 0.01 vs. UUO D3 control; ^##^
*p* < 0.01 vs. UUO D7 control. Sh, sham-operated group; U3C, UUO at day 3 group; U3Ze, zebularine-treated UUO group at day 3; U7, UUO at day 7 group; U7Ze, zebularine-treated UUO group at day 7.

**Figure 4 ijms-23-14045-f004:**
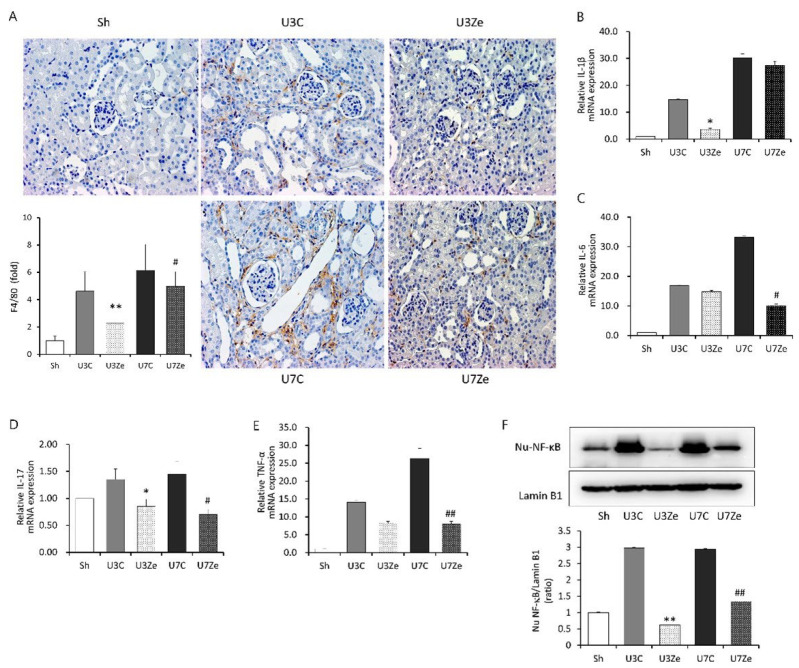
The effect of zebularine on renal inflammation after UUO. (**A**) Expression of F4/80 by immunohistochemistry mice (original magnification, ×200) and quantitative analyses in the obstructed kidneys. (**B**–**F**) QRT-PCR analysis of the obstructed kidney tissue showing upregulation of *interleukin* (*IL*)-*1β* (**B**), *IL-6* (**C**), *IL-17* (**D**), and *tumor necrosis factor* (*TNF*)-*α* (**E**), whose expression changed significantly in response to zebularine on day 3 or 7 or both after UUO. (**F**) Immunoblot analysis of nuclear NF-κB (Nu-NF-κB, p65) with quantitative analyses using Lamin B1 as a normalizer. * *p* < 0.05 vs. UUO D3 control; ** *p* < 0.01 vs. UUO D3 control; ^#^
*p* < 0.05 vs. UUO D7 control; ^##^
*p* < 0.01 vs. UUO D7 control. Sh, sham-operated group; U3C, UUO at day 3 group; U3Ze, zebularine-treated UUO group at day 3; U7, UUO at day 7 group; U7Ze, zebularine-treated UUO group at day 7.

**Figure 5 ijms-23-14045-f005:**
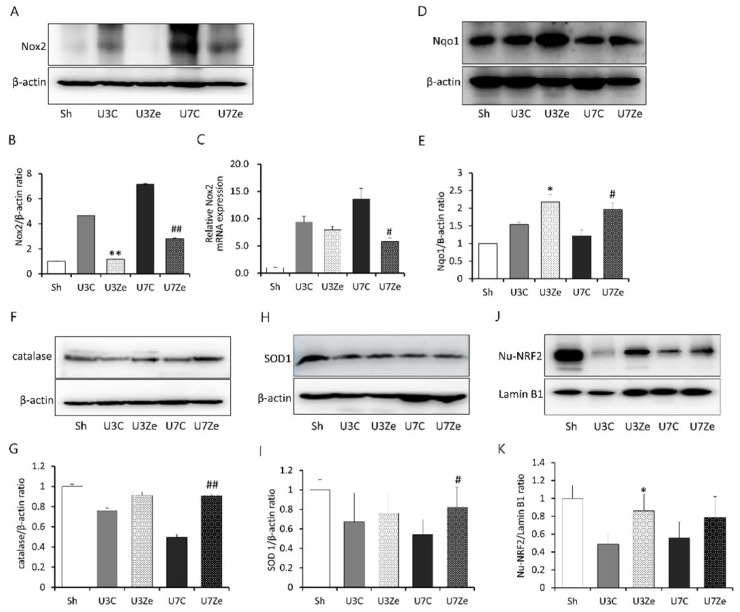
The effect of zebularine on oxidative stress-associated enzymes after UUO. (**A**–**C**) Immunoblot analysis and mRNA expression of Nox2 with quantitative analyses. Representative immunoblot image against Nox2 (**A**) and densitometric result of immunoblot images. Intensities of Nox2 bands were normalized by β-actin in the same sample (**B**). The expression level of *Nox2* mRNA was normalized by *GAPDH* mRNA in the same sample (**C**). (**D**–**I**) Western blot analysis of Nqo-1 (**D**,**E**), catalase (**F**,**G**), and SOD1 (**H**,**I**). (**J**) Immunoblot analysis for nuclear Nrf2 (Nu-NRF2) reveals that obstructive injury reduced its expression, but zebularine significantly increased its expression at all time points. (**K**) Densitometric analysis of immunoblot images against Nu-NRF2. Intensities of Nu-NRF2 bands were normalized by Lamin B1 in the same sample. * *p* < 0.05 vs. UUO D3 control; ** *p* < 0.01 vs. UUO D3 control; ^#^
*p* < 0.05 vs. UUO D7 control; ^##^
*p* < 0.01 vs. UUO D7 control. Sh, sham-operated group; U3C, UUO at day 3 group; U3Ze, zebularine-treated UUO group at day 3; U7, UUO at day 7 group; U7Ze, zebularine-treated UUO group at day 7.

**Figure 6 ijms-23-14045-f006:**
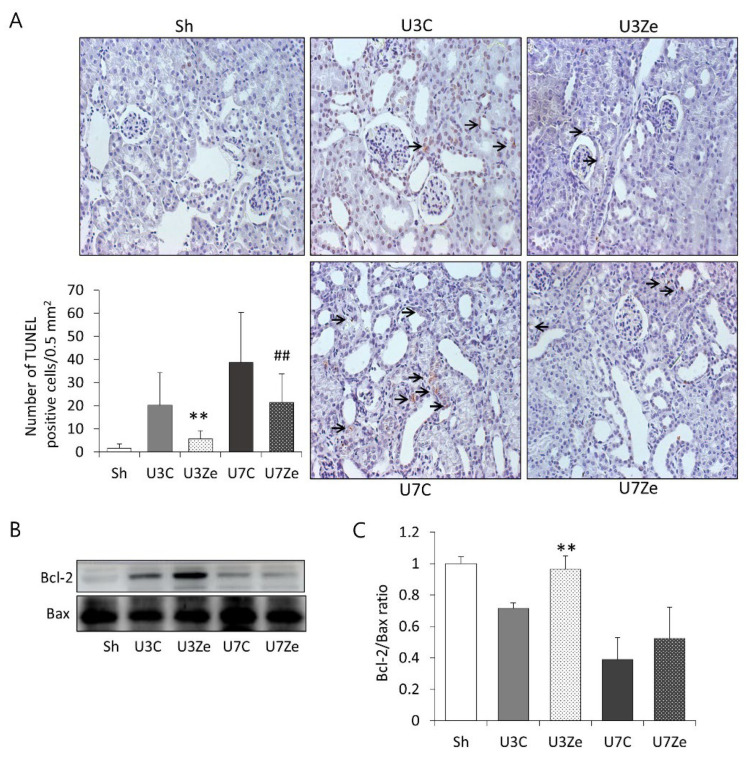
The effect of zebularine on apoptosis after UUO. (**A**) Representative renal sections stained in the TUNEL assay and quantitative analyses in the obstructed kidneys. Arrows indicate positive cells. (**B**,**C**) Immunoblot and quantitative analyses of Bcl-2 and Bax; the ratio of Bcl-2 to Bax expression decreased after UUO but increased in response to zebularine treatment. ** *p* < 0.01 vs. UUO D3 control; ^##^
*p* < 0.01 vs. UUO D7 control. Sh, sham-operated group; U3C, UUO at day 3 group; U3Ze, zebularine-treated UUO group at day 3; U7, UUO at day 7 group; U7Ze, zebularine-treated UUO group at day 7.

**Figure 7 ijms-23-14045-f007:**
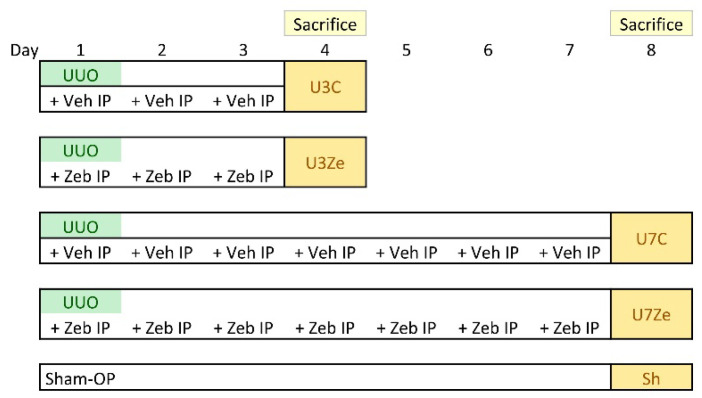
The schemes of experimental mouse groups. Zeb, zebularine; IP, intraperitoneal injection; Veh, vehicle.

## Data Availability

The data which are presented in this manuscript and support the findings of this study are available on request from the corresponding authors.

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
