# Peer review of "The Protective Effect of Zebularine, an Inhibitor of DNA Methyltransferase, on Renal Tubulointerstitial Inflammation and Fibrosis"

_ijms, 2022, doi:10.3390/ijms232214045_

Round 1

Reviewer 1 Report

The authors of the article are thanked for the quality of the work and the choice of the subject relating to the possibility of DNA methyltransferases as therapeutic targets for treating renal inflammation and fibrosis. Koh and collaborators discuss the effect of zebularine, a DNA methyltransferase inhibitor, on renal inflammation and fibrosis in the murine unilateral ureteral obstruction (UUO) model. The manuscript entitled „The Protective Effect of Zebularine, an Inhibitor of DNA Methyltransferase, on Renal Tubulointerstitial Inflammation and Fibrosis” by Koh et al, is a good study, well executed, and deserve some space in the journal. However, some concerns have been raised. My major concern is that:

Page 10 lines 307-322: The studies carried out are complicated. While reading the text, I made a diagram that helped me understand the number and names of the groups, as well as  the number of animals per group. Please add a figure (diagram, draft or schema) to the section Material and Methods, that shows the model of the experiment.

Page 10 lines 326-328: How the fibrotic area was quantified? Manually or software was used? Please explain.

Page 11 lines 335-336: How the selected fields were analysed? Manually or software was used? Please explain.

Figure 1: Immunohistochemistry without a proper negative control assume to be not convincing. Please include the blinded manner image in the plate as well (add to to Figure 1A).

Figure 3: Immunohistochemistry without a proper negative control assume to be not convincing. Please include the blinded manner image in the plate as well (add to to Figure 3A).

Figure 4: Immunohistochemistry without a proper negative control assume to be not convincing. Please include the blinded manner image in the plate as well (add to to Figure 4A).

Author Response

The authors of the article are thanked for the quality of the work and the choice of the subject relating to the possibility of DNA methyltransferases as therapeutic targets for treating renal inflammation and fibrosis. Koh and collaborators discuss the effect of zebularine, a DNA methyltransferase inhibitor, on renal inflammation and fibrosis in the murine unilateral ureteral obstruction (UUO) model. The manuscript entitled „The Protective Effect of Zebularine, an Inhibitor of DNA Methyltransferase, on Renal Tubulointerstitial Inflammation and Fibrosis” by Koh et al, is a good study, well executed, and deserve some space in the journal. However, some concerns have been raised. My major concern is that:

I sincerely appreciate your favorable assessment of our manuscript. I tried my best to resolve your valuable concerns which, I believe, contribute to the completeness of our manuscript.

Page 10 lines 307-322: The studies carried out are complicated. While reading the text, I made a diagram that helped me understand the number and names of the groups, as well as the number of animals per group. Please add a figure (diagram, draft or schema) to the section Material and Methods, that shows the model of the experiment.

I appreciate your invaluable comment. According to your comment, I add a figure that explains the procedure of animal experiments as follows.

Page 10 lines 326-328: How the fibrotic area was quantified? Manually or software was used? Please explain.

Thank you so much for your sharp pointing out. Since avoiding the observer’s intentional bias is the most critical factor in comparing the fibrosis among kidney tissues, we randomly chose and photographed manually tissue slides, and then the positive area of fibrosis was analyzed using software, ImageJ 1.49 (NIH, Bethesda, MD, USA). Briefly, we randomly photographed at least 6 fields per each kidney tissue with total 40 fields per each experimental group, and the fibrotic areas of the images were detected and quantitated by ImageJ 1.49.  

I corrected the text as follows.

More than 6 fields per each kidney tissue with total 40 fields per each experimental group were randomly chosen and photographed manually, and the fibrotic area was detected and quantified using ImageJ 1.49 software (National Institutes of Health, Bethesda, MD, USA). The difference among experimental groups was statistically verified using one-way analysis of variance with Bonferroni correction.

Page 11 lines 335-336: How the selected fields were analysed? Manually or software was used? Please explain.

Thank you so much again for your sharp pointing out. Also, in IHC experiments, we avoided the bias by random selection of images per each kidney tissues and then analysis by software ImageJ 1.49.

I corrected the text as follows. 

After counterstaining with haematoxylin, more than 6 fields per each kidney tissue with total 40 fields per each experimental group were randomly selected and photographed manually. Stain-positive cells were detected and quantitated by ImageJ software 1.49 (National Institutes of Health).

Figure 1: Immunohistochemistry without a proper negative control assume to be not convincing. Please include the blinded manner image in the plate as well (add to to Figure 1A).

I sincerely appreciate your insightful comment. I admit and regret that I omitted negative control in IHC experiments. To include the negative control for Figure 1A, we conducted IHC against U7C which showed the highest level of Dnmt1 expression, by skipping anti-Dnmt1 antibody incubation. Briefly, a section of U3C tissue were incubated in primary antibody incubating solution without anti-DNMT1 antibody. After washing in PBS, the section was incubated for 60 min with peroxidase-conjugated anti-rabbit IgG (Jackson ImmunoResearch Laboratories, West Grove, PA, USA) as a secondary antibody and then reacted with a mixture of 3,3′-diaminobenzidine (0.05%)-containing H2O2 (0.01%) for the colour reaction. After counterstaining with haematoxylin, the section was photographed as follows. The result is presented as Supplementary Figure S3.

Figure 3: Immunohistochemistry without a proper negative control assume to be not convincing. Please include the blinded manner image in the plate as well (add to to Figure 3A).

I sincerely appreciate again your insightful comment. To include the negative control for Figure 3A, we conducted IHC against U7C which showed the maximum level of a-SMA protein, by skipping anti-a-SMA antibody incubation. Briefly, a section of U7C tissue were incubated in primary antibody incubating solution without anti-a-SMA antibody. After washing in PBS, the section was incubated for 60 min with peroxidase-conjugated anti-mouse IgG (Jackson ImmunoResearch Laboratories, West Grove, PA, USA) as a secondary antibody and then reacted with a mixture of 3,3′-diaminobenzidine (0.05%)-containing H2O2 (0.01%) for the colour reaction. After counterstaining with haematoxylin, the section was photographed as follows. The result is presented as Supplementary Figure S3.

Figure 4: Immunohistochemistry without a proper negative control assume to be not convincing. Please include the blinded manner image in the plate as well (add to to Figure 4A).

I thank you so much again for your insightful comment. To include the negative control for Figure 4A, we conducted IHC against U7C which showed the highest positivity, by skipping anti-F4/80 antibody incubation. Briefly, a section of U7C were incubated in primary antibody incubating solution without anti-F4/80 antibody. After washing in PBS, the section was incubated for 60 min with peroxidase-conjugated anti-rat IgG (Jackson ImmunoResearch Laboratories, West Grove, PA, USA) as a secondary antibody and then reacted with a mixture of 3,3′-diaminobenzidine (0.05%)-containing H2O2 (0.01%) for the colour reaction. After counterstaining with haematoxylin, the section was photographed as follows. The result is presented as Supplementary Figure S3.

Reviewer 2 Report

The Protective Effect of Zebularine, an Inhibitor of DNA Methyltransferase, on Renal Tubulointerstitial Inflammation and Fibrosis

Dear Authors,

Congratulations on writing such an interesting article to examine the effect of zebularine, a DNMT inhibitor, on the progression of EMT, inflammation, and oxidative stress in murine kidneys with unilateral ureteral obstruction (UUO).

The following are my comments and suggestions:

1.      Regarding the abstract, kindly provide precious information about finding or results and mention changes in percentage or in the fold.

2.      Liver - Does any observation notice regarding liver enzyme elevation? Please describe it in detail in respect of molecular pathways.

3.      I was wondering why the author did not present or show the blood and urine report data. Kindly provide the information about it and other cytokines that get modulated during the progression of the rodent model’s protocol.

4.      Kindly provide the references in the method section where it needs to be mentioned.

5. Figures 1, 2, and 3 – please use the marker to show inflammation/fibrosis as same as figure 6. A minimum of n=4 was expected for such studies.

6.       Fig 5 – U3Ze is showing better results than other concentrations. Can the author explain the significance of these results?

7.      Kindly provide a pictorial diagram for the action of the mechanism of Zebularine against the progression of EMT, 78 inflammation, and oxidative stress.

Discussion:

Kindly provide brief details about the metabolism of Zebularine and how they get modulated or affects in case of disease conditions. Kindly explain whether the secondary metabolites originated or metabolites get affected by Zebularine. 

Kindly use a few bioinformatics tools and please predict the GO and KEGG pathways that get affected by Zebularine.

Author Response

The Protective Effect of Zebularine, an Inhibitor of DNA Methyltransferase, on Renal Tubulointerstitial Inflammation and Fibrosis

Dear Authors,

Congratulations on writing such an interesting article to examine the effect of zebularine, a DNMT inhibitor, on the progression of EMT, inflammation, and oxidative stress in murine kidneys with unilateral ureteral obstruction (UUO).

I sincerely appreciate your beneficial view of our manuscript. Your insightful comments and critics are invaluable, which will be very helpful for improving the completeness of our manuscript.

The following are my comments and suggestions:

  1. Regarding the abstract, kindly provide precious information about finding or results and mention changes in percentage or in the fold.

I sincerely appreciate your valuable comment. Following your suggestion, I modified the abstract by adding quantitative data as follows.

Zebularine decreased trichrome, α-smooth muscle actin, collagen IV, and transforming growth factor-β1 staining by 56.2%. 21.3%, 30.3%, and 29.9% respectively at 3 days, and by 54.6%, 41.9%, 45.9%, and 61.7%, respectively at 7 days after UUO. Zebularine downregulated mRNA expression levels of matrix metalloproteinase (MMP)-2, MMP-9, fibronectin, and Snail1 by 48.6%. 71.4%, 31.8%, and 42.4% respectively at 7 days after UUO. Zebularine also suppressed nuclear factor-κB (NF-κB) and pro-inflammatory cytokines such as tumor necrosis factor-α, interleukin (IL)-1β, and IL-6 by 69.8%, 74.9%, and 69.6%, respectively, in obstructed kidneys as well. Furthermore, inhibiting DNA methyltransferase buttressed the nuclear expression of nuclear factor (erythroid-derived 2)-like factor 2, which upregulated downstream effectors such as catalase (1.838-fold increase at 7 days, P < 0.01), superoxide dismutase 1 (1.494-fold increase at 7 days, P < 0.05), and NAD(P)H: quinone oxidoreduate-1 (1.376-fold increase at 7 days, P < 0.05) in obstructed kidneys.

  1. Liver - Does any observation notice regarding liver enzyme elevation? Please describe it in detail in respect of molecular pathways.

I was impressed by your insightful comment and question. I did not think of the effect of zebularine on the liver. Hearing this question, I thought I had to check the liver enzymes, since almost all chemicals are metabolized through the liver. First of all, I feel sorry that I have no choice but to answer your question by researching the references.

As you suggested, zebularine is metabolized in the liver. Zebularine is first-pass converted to uridine by aldehyde oxidase (AO) which is abundant in hepatocyte cytosol [1]. Since the first-pass metabolic rate of zebularine is somewhat rapid, its oral bioavailability seems to be limited and safe with less side effects than that of other demethylating agents such as 5-azacytidine and 5-azadeoxycytidine [2]. To be consistently, mice were well tolerable to zebularine administered by intraperitoneal or intravenous injection and showed any signs of liver and kidney injury [3,4,5]. Moreover, diabetic rats treated intraperitoneally with zebularine at a dose of 225 mg/kg/day, daily for 14 days were tolerable and prolonged survival of pancreatic islet allotransplants until 90 days [6]. Therefore, mice in this study treated with zebularine at a dose of 225 mg/kg/day, daily for 3 or 7 days should be well tolerable with no hepatic and kidney injuries.

However, since administration of zebularine by daily intravenous injection at a dose of 250 mg/kg/day for 10 days with 2-day interval induced modest increase of hepatic enzyme ALT in cynomolgus monkeys [7], it should be noted that safety of zebularine treatment is variable among species.

Once again, I thank you so much for your wonderful question and I inserted new sentences regarding the safety of zebularine treatment as follows.

The first limitation of this study is that the side effect of zebularine such as hepatic injury was not measured. The first-pass metabolism of zebularine occurs in the liver. Zebularine is metabolized to uridine by aldehyde oxidase (AO) which is abundant in hepatocyte cytosol [1]. Since the first-pass metabolic rate of zebularine is somewhat rapid, its oral bioavailability seems to be limited and safe with less side effects than that of other demethylating agents such as 5-azacytidine and 5-azadeoxycytidine [2]. To be consistently, mice were well tolerable to zebularine administered by intraperitoneally or intravenously and showed no sign of liver and kidney injury [3,4,5]. Moreover, diabetic rats treated intraperitoneally with zebularine at a dose of 225 mg/kg/day, daily for 14 days were tolerable and prolonged survival of pancreatic islet allotransplants until 90 days [6]. Therefore, mice in this study treated with zebularine at a dose of 225 mg/kg/day, daily for 3 or 7 days should be well tolerable with no hepatic and kidney injuries. However, since administration of zebularine by daily intravenous injection at a dose of 250 mg/kg/day for 10 days with 2-day interval induced modest increase of hepatic enzyme ALT in cynomolgus monkeys [7], it should be cautious that safety of zebularine treatment is variable among species.

[1] Klecker, R.W.; Cysyk, R.L.; Collins, J.M. Zebularine metabolism by aldehyde oxidase in hepatic cytosol from humans, monkeys, dogs, rats, and mice: Influence of sex and inhibitors. Bioorg. Med. Chem. 2006, 14, 62-66.

[2] Holleran, J.L.; Parise, R.A.; Joseph, E.; Eiseman, J.L.; Covey, J.M.; Glaze, E.R.; Lyubimov, A.V.; Chen, Y.-F.; D'Argenio, D.Z.; Egorin, M.J. Plasma pharmacokinetics, oral bioavailability, and interspeciesscaling of the DNA methyltransferase inhibitor, zebularine. Clin. Cancer Res. 2005,11(10), 3862-3868.

[3] Chen, M.; Shabashvili, D.; Nawab, A.; Yang, S.X.; Dyer, L.M.; Brown, K.D.; Hollingshead, M.;  Hunter, K.W.; Kaye, F.J.; Hochwald, S.N.; Marquez, V.E.; Steeg, P.; Zajac-Kaye, M. DNA methyltransferase inhibitor, zebularine, delays tumor growth and induces apoptosis in a genetically engineered mouse model of breast cancer. Molecular Cancer Therapeutics 2012,11, 370–382.

[4] Herranz, M.; Martin-Caballero, J.; Fraga, M.F.; Ruiz-Cabello, J.; Flores, J.M.; Desco, M.; Victor, M.; Esteller, M. The novel DNA methylation inhibitor zebularine is effective against the development of murine T-cell lymphoma. Blood 2006, 107, 1174–1177.

[5] Yoo, C.B.; Chuang, J.C.; Byun, H.M.; Egger, G.; Yang, A.S.; Dubeau, L.; Long, T.; Laird, P.W.; Marquez, V.E.; Jones, P.A. Long-term epigenetic therapy with oral zebularine has minimal side effects and prevents intestinal tumors in mice. Cancer Prev. Res. 2008, 1(4), 233-40.

[6] Nittby, H.; Ericsson, P.; Förnvik, K.; Strömblad, S.; Jansson, L.; Xue, Z.; Skagerberg, G.; Widegren, B.; Sjögren, H.O.; Salford, L.G. Zebularine induces long-term survival of pancreatic islet allotransplants in streptozotocin treated diabetic rats. PLOS One 2013, 8(8), e71981.

[7] Johnson, W.D.; Harder, J.B.; Naylor, J.; McCormick, D.L.; Detrisac, C.J.; Glaze, E.R.; Tomaszewski, J.E. A pharmacokinetic/pharmacodynamic approach to evaluating the safety of zebularine in non-human primates. Proc. Amer. Assoc. Cancer Res. 2006, 47. 

  1. I was wondering why the author did not present or show the blood and urine report data. Kindly provide the information about it and other cytokines that get modulated during the progression of the rodent model’s protocol.

I appreciate your very sharp pointing out. Unfortunately, in this study, we did not measure the blood urea nitrogen (BUN) or serum creatinine as renal functional parameters. Since many previous studies have reported that BUN or serum creatinine was not significantly changed by UUO because a contralateral kidney has good function [1,2], and usually compensates for the decreased function of UUO kidney [3], BUN and serum creatinine do not seem to precisely reflect the renal function in an animal model of UUO [4]. That’s why we usually check the renal function parameters in UUO model. I feel very sorry that I can not show you the effect of zebularine on the renal function. But I would like to report the renal function with the involved mechanism in this model in near future.

[1] Ning, X.H.; Ge, X.F.; Cui, Y.; An, H.X. Ulinastatin inhibits unilateral ureteral obstruction-induced renal interstitial fibrosis in rats via transforming growth factor β (TGF-β)/Smad signalling pathways. Int. Immunopharmacol 2013, 15, 406-413.

[2] Wu, W.P.; Chang, C.H.; Chiu, Y.T.; Ku, C.L.; Wen, M.C.; Shu, K.H.; Wu, M.J. A reduction of unilateral ureteral obstruction-induced renal fibrosis by a therapy combining valsartan with aliskiren. Am. J. Physiol. Renal Physiol. 2010, 299, F929-F941.

[3] Cai, X.R.; Zhou, Q.-C.; Yu, J.; Feng, Y.-Z.; Xian, Z.-H.; Yang, W.C.; Mo, X.-K. Assessment of Renal Function in Patients with Unilateral Ureteral Obstruction Using Whole-Organ Perfusion Imaging with 320-

Detector Row Computed Tomography. PLOS One 2015, 10(4), e0122454

[4] Chung, S.; Yoon, H.E.; Kim, S.J.; Kim, S.J.; Koh, E.S.; Hong, Y.A.; Park, C.W.; Chang, Y.S.; Shin, S.J. Oleanolic acid attenuates renal fibrosis in mice with unilateral ureteral obstruction via facilitating nuclear translocation of Nrf2. Nutr. Metab. 2014, 11(1), 2.

I inserted new text as follows.

The second limitation of this study is that the effect of zebularine on the renal function was not analyzed. Since many previous studies have reported that BUN or serum creatinine was not significantly changed by UUO because a contralateral kidney has good function [1,2], and usually compensates for the decreased function of UUO kidney [3], BUN and serum creatinine do not seem to precisely reflect the renal function in an animal model of UUO [4]. So, it is expected that the effect of zebularine on the renal function may be obvious in a model where both kidneys undergo fibrosis.

In UUO model, tubular damage, inflammation, and fibrosis events take place in a very complicated network [1]. Although I am not sure if the chronological order of these events is meaningful, the temporal sequence of these events is tubular damage, inflammation, and the development of fibrosis. So, in UUO model, inflammatory cytokines increase, which is followed by the expression of anti-inflammatory or pro-fibrotic cytokines. In this study, zebularine inhibits the pro-fibrotic or anti-inflammatory cytokines such as TGF-b, IL-10 and IL-11 (Supplementary Figure S1) as well as pro-inflammatory cytokines such as TNF-a, IL-1b, IL-6, and IL-17. So, both pro-inflammatory and anti-inflammatory cytokines may be targets of zebularine and the way how zebularine affects the expression of cytokines will be mechanistically specific.

[1] Ucero, A.C.; Benito-Martin, A.; Izquierdo, M.C.; Sanchez-Niño, M.D.; Sanz, A.B.; Ramos, A.M.; Berzal, S.; Ruiz-Ortega, M.; Egido, J.; Ortiz, A. Unilateral ureteral obstruction: beyond obstruction. Int. Urol. Nephrol. 2014, 46(4), 765-776.

I modified the text as follows.

Regarding inflammation, zebularine treatment alleviated the activation of NF-κB and the subsequent expression of inflammatory cytokines in obstructed kidneys (Figure 4). In the UUO model, inflammatory cytokines increase, which is followed by the expression of anti-inflammatory or pro-fibrotic cytokines [19]. The intriguing part of the results is that zebularine also attenuated anti-inflammatory molecules, such as IL-10 and IL-11, as well as pro-inflammatory cytokines (supplementary Figure S1). So, it can be inferred that DNMT may affect the expression of most cytokines regardless of their characteristics and roles, and both pro-inflammatory and anti-inflammatory cytokines may be targets of zebularine which prevents the expression of cytokines in a mechanistically specific manner. Given that the final effect of inhibiting DNMT during renal fibrosis was more favorable for renoprotection, DNMT appears to be more influential toward pro-inflammatory than anti-inflammatory conditions in the mouse UUO model. Accordingly, this observation suggests that inhibiting DNMT in renal fibrosis could be a therapeutic strategy to correct the imbalance between pro-inflammatory and anti-inflammatory molecules.

  1. Kindly provide the references in the method section where it needs to be mentioned.

I sincerely appreciate your considerate comment. Following your suggestion, I added four references.

Histology and Immunohistochemistry

  1. Chung, S.; Son, M.; Chae, Y.; Oh, S.; Koh, E.S.; Kim, Y.K.; Shin, S.J.; Park, C.W.; Jung, S.C.; Kim, H.-S. Fabry disease exacerbates renal interstitial fibrosis after unilateral ureteral obstruction via impaired autophagy and enhanced apoptosis. Kidney Res. Clin. Pract. 2020, 40(2), 208-219.
  2. Chung, S.; Kim, S.; Son, M.; Kim, M.; Koh, E.S.; Shin, S.J.; Park, C.W.; Kim, H.-S. Inhibition of p300/CBP-Associated Factor Attenuates Renal Tubulointerstitial Fibrosis through Modulation of NF-kB and Nrf2. J. Mol. Sci. 2019, 20(7), 1554.

Renal Collagen Content Assay

  1. Chung, S.; Kim, S.; Kim, M.; Koh, E.S.; Yoon, H.E.; Kim, H.S.; Park, C.W.; Chang, Y.S.; Shin, S.J. T-type calcium channel blocker attenuates unilateral ureteral obstruction-induced renal interstitial fibrosis by activating the Nrf2 antioxidant pathway. J. Transl. Res. 2016, 8, 4574–4585.
  2. Chung, S.; Yoon, H.E.; Kim, S.J.; Kim, S.J.; Koh, E.S.; Hong, Y.A.; Park, C.W.; Chang, Y.S.; Shin, S.J. Oleanolic acid attenuates renal fibrosis in mice with unilateral ureteral obstruction via facilitating nuclear translocation of Nrf2. Metab. 2014, 11, 2.

I believe that the addition of new references will inform the readers the experimental method in detail, which is the important purpose of this journal. I thank you so much for your considerate review again!

  1. Figures 1, 2, and 3 – please use the marker to show inflammation/fibrosis as same as figure 6. A minimum of n=4 was expected for such studies.

I sincerely appreciate your careful examination. Following your suggestion, I added the marker in Figure 1A as follows. But, unfortunately, the positive area of Figure 2A and Figure 3A is too wide to indicate using the marker. I feel sorry that I cannot use the marker and will figure out to present the positive area more clearly in the future.

The tissues were from 6 mice and to get the result of the immunohistochemistry, we analyzed 40 images at least per each experimental group.

  1. Fig 5 – U3Zeis showing better results than other concentrations. Can the author explain the significance of these results?

I sincerely appreciate your wonderful question which is very impressive and insightful. Figure 5 is the result of the effect of zebularine on the oxidative stress in UUO model. Although we did not analyze the mechanisms involved in zebularine-modulated Nox2 expression yet, zebularine almost completely prevented up-regulation of Nox2 expression at 3 days in obstructed kidneys. In addition, decrease of nuclear Nrf2 and increase of Nqo1 which are major players defensing against oxidative stress are prominent at 3 days post-UUO. However, as oxidative stress sustains beyond 3 days until 7days, the effect of zebularine seems to weaken gradually. Therefore, we speculate that U3Ze showed better effect than U7Ze.  

I inserted the text as follows in Figure 5.

Interestingly, U3Ze showed an inhibitory effect more substantial than that of U7Ze. Zebularine almost completely prevented up-regulation of Nox2 expression at 3 days in obstructed kidneys. In addition, the decrease of nuclear Nrf2 and the increase of Nqo1 are prominent at 3 days post-UUO. However, as oxidative stress sustains beyond 3 days until 7 days, the effect of zebularine seems to weaken gradually. Therefore, it was likely that U3Ze showed a better effect than U7Ze.

  1. Kindly provide a pictorial diagram for the action of the mechanism of Zebularine against the progression of EMT, 78 inflammation, and oxidative stress.

 Thank you so much for your considerate examination. Following your suggestion, we made a pictorial diagram as follows. The three major targets of zebularine in this model may be Nrf2, NF-kB, and Nox2. By modulating these molecules, zebularine suppresses oxidative stress, inflammation, and EMT, leading to prevention of fibrosis in UUO.

I revised the text as follows.

The three major targets of zebularine in the UUO model may be Nrf2, NF-kB, and Nox2. By modulating these molecules, zebularine suppresses oxidative stress, inflammation, and EMT, leading to prevention of fibrosis in UUO (supplementary Figure S2).

Discussion:

Kindly provide brief details about the metabolism of Zebularine and how they get modulated or affects in case of disease conditions. Kindly explain whether the secondary metabolites originated or metabolites get affected by Zebularine. 

I sincerely appreciate your insightful question which improves the completeness of our manuscript.

It is known that zebularine is metabolized sequentially to uridine, uracil, dihydrouracil, b-ureiodopropionic acid and b-alanine as shown in a figure below [1]. The first-pass metabolism of zebularine to uridine takes place in hepatocyte cytosol by aldehyde oxidase. Since the half-life of zebularine is short, its bioavailability is limited and safe to animals with less side effects. Although pyrimidine analogs have hepatotoxic effects, metabolites of zebularine are indistinguishable from endogenous ones and doesn’t seem to have toxic effects. It was reported that in patients with diabetic kidney disease, dihydrouracil and b -ureidopropionic acid are significantly down-regulated, which leads to decrease of pantothenate and CoA biosynthesis [2]. Therefore, it can be speculated that zebularine might have the therapeutic potential in diabetic kidney disease.

[1] Holleran, J.L.; Parise, R.A.; Joseph, E.; Eiseman, J.L.; Covey, J.M.; Glaze, E.R.; Lyubimov, A.V.; Chen, Y.-F.; D'Argenio, D.Z.; Egorin, M.J. Plasma pharmacokinetics, oral bioavailability, and interspeciesscaling of the DNA methyltransferase inhibitor, zebularine. Clin. Cancer Res. 2005,11(10), 3862-3868.

[2] Ma, T.; Liu, T.; Xie, P.; Jiang, S.; Yi, W.; Dai, P.; Guo, X. UPLC-MS-based urine nontargeted metabolic profiling identifies dysregulation of pantothenate and CoA biosynthesis pathway in diabetic kidney disease. Life Sci. 2020, 258, 118160.

I added new text as follows.

The third limitation of this study is the effect of metabolites of zebularine was not analyzed. It is known that zebularine is metabolized sequentially to uridine, uracil, dihydrouracil, b-ureiodopropionic acid and b-alanine [39]. Since the half-life of zebularine is short, its bioavailability is limited and safe to animals with less side effects. Although pyrimidine analogs have hepatotoxic effects, metabolites of zebularine are indistinguishable from endogenous ones and don’t seem to have toxic effects. It was reported that in patients with diabetic kidney disease, dihydrouracil and b-ureidopropionic acid are significantly down-regulated, which leads to a decrease of pantothenate and CoA biosynthesis [48]. Therefore, it can be speculated that zebularine might have therapeutic potential in diabetic kidney disease.

Kindly use a few bioinformatics tools and please predict the GO and KEGG pathways that get affected by Zebularine.

I sincerely appreciate your future-oriented question. Since the effect of zebularine on fibrosis model including UUO model was not yet analyzed by next-generation sequencing, we really wanted to apply NGS in this model. However, due to the shortage of our research budget, we could not perform NGS in this study.

As far as I know, one study reported DNA microarray result of zebularine-treated multiple myeloma cells [1], and another study reported the RNA-seq result of nucleus accumbens of zebularine-treated mice [2]. But, unfortunately, in those results the expression of genes involved in oxidative stress, inflammation, and fibrosis was not significantly changed by zebularine. I hope that I will be able to perform RNA sequencing in this model and publish the result.

[1] Pompeia, C.; Hodge, D.R.; Plass, C.; Wu, Y.Z.; Marquez, V.E.; Kelley, J.A.; Farrar, W.L. Microarray analysis of epigenetic silencing of gene expression in the KAS-6/1 multiple myeloma cell line. Cancer Res. 2004, 64(10), 3465-3473.

[2] Hao, B.; Fan, B.-F.; Cao, C.-C.; Liu, L.; Xuan, S.-M.; Wang, L.; Gao, Z.-J.; Shi, Y.-W.; Wang, X.-G.; Zhao, H. Genes and pathways associated with fear discrimination identified by genome-wide DNA methylation and RNA-seq analyses in nucleus accumbens in mice.  Prog. Neuropsychopharmacol. Biol. Psychiatry 2023, 120, 110643.
